# Mandibular Torus as a New Index of Success for Mandibular Advancement Devices

**DOI:** 10.3390/ijerph192114154

**Published:** 2022-10-29

**Authors:** Teresa Diaz de Teran, Pedro Muñoz, Felix de Carlos, Emilio Macias, Marta Cabello, Olga Cantalejo, Paolo Banfi, Antonello Nicolini, Paolo Solidoro, Monica Gonzalez

**Affiliations:** 1Sleep and Ventilation Unit, Pneumology Department, Hospital Universitario Marqués de Valdecilla, 39008 Santander, Spain; 2Cantabria Primary Health Care Management, Instituto Marqués de Valdecilla (IDIVAL), 39011 Santander, Spain; 3Department of Surgery and Medical-Surgical Specialties, Area of Orthodontics, Faculty of Medicine, University of Oviedo, 33003 Oviedo, Spain; 4Don Gnocchi Foundation IRCCS, 20121 Milan, Italy; 5Division of Respiratory Diseases Cardiovascular and Thoracic Department, AOU Città della Salute e della Scienza di Torino, 10126 Torino, Italy; 6Instituto Marqués de Valdecilla (IDIVAL), University of Cantabria, 39011 Santander, Spain

**Keywords:** obstructive sleep apnoea, mandibular advancement devices, cephalometry, mandibular torus

## Abstract

Background: In obstructive sleep apnoea (OSA), treatment with mandibular advancement devices (MADs) reduces patients’ Apnoea–Hypopnoea index (AHI) scores and improves their sleepiness and quality of life. MADs are non-invasive alternatives for patients who cannot tolerate traditional continuous positive airway pressure (CPAP) therapy. The variability of responses to these devices makes it necessary to search for predictors of success. The aim of our study was to evaluate the presence of mandibular torus as a predictor of MAD efficacy in OSA and to identify other potential cephalometric factors that could influence the response to treatment. Methods: This was a retrospective cohort study. The study included 103 patients diagnosed of OSA who met the criteria for initiation of treatment with MAD. Structural variables were collected (cephalometric and the presence or absence of mandibular torus). Statistical analysis was performed to evaluate the existence of predictive factors for the efficacy of MADs. Results: A total of 103 patients who were consecutively referred for treatment with MAD were included (89.3% men); the mean age of the participants was 46.3 years, and the mean AHI before MAD was 31.4 (SD 16.2) and post- MAD 11.3 (SD 9.2). Thirty-three percent of patients had mandibular torus. Torus was associated with a better response (odds ratio (OR) = 2.854 (*p* = 0.035)) after adjustment for sex, age, body mass index (BMI; kg/m^2^), the angle formed by the occlusal plane to the sella–nasion plane (OCC plane to SN), overinjection, and smoking. No cephalometric predictors of efficacy were found that were predictive of MAD treatment success. Conclusions: The presence of a mandibular torus practically triples the probability of MAD success. This is the simplest examination with the greatest benefits in terms of the efficacy of MAD treatment for OSA.

## 1. Introduction

Continuous positive airway pressure is the most widely used treatment and an efficacious option for patients with obstructive sleep apnoea (OSA) [1]. On the other hand, the effectiveness of mandibular advancement devices (MADs) as valid non-CPAP (continuous positive airway pressure (CPAP)) alternatives has been demonstrated via solid scientific evidence [2]. Some authors suggest that movement of the mandible using MADs achieves anterior repositioning of the tongue and soft palate, as well as changes in the morphology and volume of the upper airway (UA). This new situation increases motor muscle tone and reduces UA collapsibility while inducing changes in local pharyngeal pressures, which contribute to normalising the physiological properties of the UA [3,4].

The variability of the responses to these devices reported in the scientific literature makes it necessary to search for predictors of success that allow us to choose the best candidates. Two systematic reviews [5,6] that attempted to evaluate the different clinical, diagnostic, and cephalometric factors involved in the response to MADs once again highlighted the heterogeneity of the studies and the absence of exportable conclusions.

The mandibular torus is an exostosis located inside the mandible under the lingual surface [7] (the bony prominence that usually occurs at the level of the premolars in the internal table of the lower jaw in some individuals). Most appear bilaterally and may have varying degrees in size depending on the population studied [8,9]. Palm et al. [10] related the existence of the mandibular torus to the success of a MAD treatment. From a clinical point of view, this finding is relevant because its presence is easily detectable by simple palpation and/or visualisation. 

The use of cephalometric measurements in multiple studies has also been attempted in order to identify the aspects that should be considered as predictors of MAD efficacy. In the previously cited review [6], only 13 articles could be considered given the methodological characteristics of the studies, and it is currently not possible to determine which measurements reliably determine a patient’s response to a MAD treatment. Furthermore, the performance of a correct cephalometric assessment requires, in addition to the use of a radiological technique, the need for a precise assessment of the measurements by experts. This complexity highlights the need to find a simple, low-cost examination applicable to clinical practice that can help in therapeutic decision-making with respect to MADs.

The aim of our study was to demonstrate that the presence of a mandibular torus, as determined via a routine examination in a doctor’s office, can be a predictive factor of a good response to MADs. The secondary objective was to identify factors affecting the success of treatments with MADs using cephalometric measurements. 

## 2. Material and Methods

This was a cohort study performed with a prospective collection of variables [11]. Patients referred to the Sleep and Ventilation Unit of the Hospital Universitario Marqués de Valdecilla for suspected OSA who met the following criteria for treatment with MAD were included: ≥18 years and:A diagnosis of mild-to-moderate OSA (Apnoea–Hypopnoea index (AHI) ≥ 15 and <30) with relevant symptomatology such as presence of snoring, witnessed apnoea, excessive daytime sleepiness, and asthenia not explained by other causes.A diagnosis of severe OSA (AHI ≥ 30) with CPAP rejected as the first therapeutic option.Patients with OSA undergoing CPAP treatment with a lack of adherence and/or intolerance or refusal of therapy.

Patients with severe somnolence while driving, for whom a delay in treatment with MAD represented a danger, were excluded. Other exclusion criteria included patients who would be intolerant of MADs from an orthodontic standpoint following an assessment by the prescribing pulmonologist of the Sleep Unit; lack of teeth (>8) preventing the correct anchorage of the MAD or poor mandibular advancement. We also excluded patients with serious concomitant pathologies or situations of clinical instability that, in the opinion of the prescribing physician, did not require treatment with MADs. Patients with central apnoea syndrome were excluded from this study.

According to the characteristics of each patient and following the action protocol of the Sleep and Ventilation Unit, respiratory polygraphy from the Philips Respironics Alice PDx was requested for each patient. Events were scored according to the American Academy of Sleep Medicine criteria (AASM) [12]. All patients underwent an orthopantomography and lateral neck X-ray for subsequent assessment by the orthodontist according to our Sleep Unit protocol. If there was no contraindication identified by the orthodontist and once informed consent was signed, the MAD treatment was performed. The MAD used was a two-piece custom-made adjustable device (Silensor^®^), and its design features and effectiveness have been previously published [13]. Titration occurred over a period of approximately six weeks, during which the appliance was incrementally advanced until the maximum comfortable limit of mandibular advancement was reached. Subsequently, a second sleep study was conducted with the device within a maximum period of 6 months.

The dimensional and dentofacial parameters of the airway were measured using a digital cephalometric analysis program (Dolphin Imaging Cephalometric and Tracing 10.0, Chatsworth, CA, USA). 

Data on the following variables were collected: age (years), sex, BMI (kg/m^2^), smoking history (current, past, or non-smoker), and sleepiness (excessive somnolence measured using the Epworth score (ES)). The polygraphic variables were total recording time in minutes, respiratory events measured by the apnoea/hypopnea index (AHI), mean oxyhemoglobin arterial saturation (SaO_2_), percentage of recording time with SaO_2_ <90% (CT90: <30), and desaturation index values (ODI: number of desaturations per hour of recording). 

Twenty-one cephalometric variables were recorded (see Figure 1 and Table 1). The presence of mandibular torus (bilateral exostosis along the lingual aspect of the mandible above the mylohyoid line) was assessed by inspection and/or palpation of the maxillary lower jaw internal part (see Figure 2).

This study was approved by the Medical Investigations Ethics Committee (MIEC) of Cantabria (Reference 2013.171). Written informed consent was obtained from all the patients. All data were obtained in compliance with Spanish laws and the guidelines of the Declaration of Helsinki.

### 2.1. Defining MAD Effectiveness

To consider the efficacy of the MAD, we defined the patients’ responses to the MAD treatment as:Effective: defined as an AHI < 10 with decrease of more than 50% from baseline [14,15].Partially effective: defined as an AHI ≥ 10 with a decrease of more than 50% from baseline.Not effective: defined as an AHI decrease lower than 50% from baseline, regardless of the current AHI.

### 2.2. Statistical Analysis

Descriptive analysis of the sample was performed using absolute and relative frequency distributions for qualitative variables. For quantitative variables, the fit of the data to a normal distribution was checked using the Kolmogorov–Smirnov test by using the mean and standard deviation or the median and interquartile range, depending on the case.

Chi-square tests were used to examine the relationships between the qualitative variables. Student’s *t*-test was used to compare paired samples, and non-parametric tests were used if these conditions were not met. If the variables were normally distributed, the non-parametric Mann–Whitney *U* test was used for independent samples, and the Wilcoxon test was used if two variables were related. When three or more variables were analysed, we used the non-parametric Kruskal–Wallis test for independent samples. A value of *p* < 0.05 was considered significant. 

A logistic regression analysis was performed. For the selection of variables for the multivariate analysis, a univariate analysis was performed with each of the variables, following the model proposed by Hosmer and Lemeshow [11]: variables with a significance of less than 0.25 were considered, and their clinical relevance was also considered, regardless of their statistical significance. To identify the existence of collinearity between the variables included in the model, the tolerance statistic [16] was used: values below 0.10 were considered to have significant collinearity, and those below 0.20 were considered to have collinearity that may be of concern.

To search for efficacy factors and construct a predictive model, the three response categories (effective, partially effective, and non-effective) were simplified to a dichotomous model (effective and non-effective), with partially effective responses grouped with the non-effective responses.

All calculations were performed using the SPSS statistical package (IBM Corp. Released 2017. IBM SPSS Statistics for Windows, Version 25.0, Armok, NY, USA).

## 3. Results

A total of 123 patients started the protocol, of which 103 completed the study. Twenty patients (16%) were excluded after the radiological tests were performed and following examinations by dentists. The main reasons for exclusion were dental problems and the risk of occlusal changes (see patient flow chart Figure 3).

The average age of the patients was 46.3 years, the majority were male (89.3%), and approximately 30% were smokers. The average body mass index was 28.6 kg/m^2^ and torus was detected in 34 patients (33%). Table 2a presents the primary characteristics of the 103 patients. Table 2b shows the characteristics according to the presence or absence of torus, finding no differences between the two groups.

The mean mandibular advancement was 68% (SD: 6.1) of maximum breakthrough and mean device compliance was 5.87 (SD 1.36) hours/night. 

The overall changes in respiratory variables pre and post-MAD, depending on the presence or absence of torus, are shown in Table 3. Significant improvement was observed in all variables, with the exception of CT90% (only in no tours group) and mean% saturation (in both groups).

The effectiveness of MAD was approximately 58%, but when grouped with the partially effective responses, the effectiveness reached up to 70% (see Table 4). While this data is of great clinical relevance, in order to search for predictive factors, the group partially effective was included in the not effective group.

In the univariate analysis (Table 5), only the presence of mandibular torus was statistically significant (OR = 3.54, 95% CI: 1.408–8.919; *p* = 0.007). Among the cephalometric variables, those that obtained *p* < 0.25 were OCC, overjet, BMI grade, smoking, age, and the existence of mandibular torus. These six variables and sex were selected for the multivariate analysis.

No collinearity was found between the variables.

Among the variables, those that obtained *p* < 0.25 were OCC, overjet, BMI, smoking, age, and the existence of mandibular torus. These six variables and sex were selected for the multivariate analysis.

Table 6 shows model 1, which includes the seven variables mentioned above, showing that torus was the only variable associated with MAD treatment success (*p* = 0.0.35; OR = 2.85).

With this model, an area under the curve of 0.706 was obtained (Figure 4).

We calculated the sensitivity, specificity, positive predictive value (PPV), and negative predictive value (NPV) to be 41.8%, 80.0%, 74.2%, and 50.0% respectively. 

The relationship between the presence of a mandibular torus and different cephalometric variables was also analysed (Table 7). The existence of a mandibular torus was associated with a greater hyoid retroposition (reduced C3-hyoid distance; Figure 5), with *p* = 0.055 (at the limit of statistical significance).

## 4. Discussion

The effectiveness of mandibular advancement devices has been previously demonstrated. Traditionally, they have been recommended for mild or moderate OSA or for those who do not tolerate CPAP treatment [17]. In adult patients with OSA, the use of CPAP is superior to that of MADs (however, this is a conditional recommendation with very low quality of evidence). In the same vein, the criterion has been maintained that OSA patients with lower AHI severity scores have better responses to these devices [18,19]. However, this has not been systematically evaluated, as most studies have included heterogeneous groups with patients with mild or moderate AHI severity scores. Some studies [20] still excluded patients with AHI values > 30 events/h from their investigations; nevertheless, recent publications have considered this patient profile. However, other studies included patients with severe OSA [21,22]. In our study, all patients had moderate or severe OSA (mean AHI: 31.4; SD: 16.2) and had a good response to the MAD treatment, with a mean decrease of 64% in the frequency of respiratory events, results that are in line with the data reported in the literature [23,24]. 

Historically, the best responders to MAD have been young patients [18,25,26], female patients, and those with lower BMIs, smaller neck circumferences [27], and supine-dependent respiratory events [28]. Although the results of our study support the above conclusions, our results did not achieve statistical significance. We did not observe a worse therapeutic response with increasing age. To confirm this, despite the fact that the mean age in our sample was 46 years, we performed a regression analysis by age and observed totally heterogeneous and non-linear results. As for sex, no conclusive results were obtained given the small number of females included in the study (barely 10.7%). This is unfortunately repeated in most publications, making it clear that we need studies that include more females. 

On the other hand, given this problem of classification into responders and non-responders to MAD, some authors propose the introduction of an easy-to-make and low-cost trial device into the therapeutic pathway of OSA patients can circumvent the problem of individual responses to treatment by allowing effective classification of patients [29].

Regarding cephalometric measurements in the upper airway, there is no established protocol, which makes it difficult to compare data between different studies.

The cephalometric variables reported in the literature in responders are decreased mandibular plane angles, reduced hyoid to mandibular plane distances, short soft pal-ates, increased parapharyngeal spaces, and increased inferior intercanine widths [30]. In a prospective study of 40 patients [31] with mild to moderate OSA, responders were found to be patients with less oropharyngeal disorders, enlargement of the upper pharyngeal space, and with decreases in the space underneath the enlargement of the intercanine mandibular width; however, none of these factors were found to be predictive of responses to MADs when the multivariate analysis was performed. A systematic review by Guarda–Nardini [6] showed heterogeneity of the parameters evaluated in the literature. Furthermore, most studies were retrospective and had small sample sizes. In our study, we were unable to demonstrate any statistically significant cephalometric predictors of efficacy. It is true that these differences with respect to the rest of the studies may be due to the different methodologies employed (in our case, inclusion of more severe OSA patients). Finally, none of the studies in the literature correlated the predictors of cephalometric efficacy with the presence of a mandibular torus. This variable (presence of a torus) has not been collected together with cephalometric variables in previous studies, and it is the most important structural variable in our study.

Thirty-three percent of the 103 patients in this study had a mandibular torus, which coincides with the prevalence reported in other studies [32]. Clinically, the fact that the presence of a torus is found to be a determining factor in the success of MAD is significant, especially taking into account the simplicity of its exploration. Other studies [10] have found that torus size is associated with the frequency of respiratory disturbances during sleep, with the smallest torus sizes seen in patients with severe OSA. Treatment success with an oral appliance occurs more frequently in patients with a larger torus than in those with no torus or a small torus. In our study, we limited our analysis by assessing its presence rather than its size. 

Mandibular tori have been postulated to be anatomical features with two different meanings in patients with OSA. On the one hand, when they are large [33], they may play a role in the aetiology of OSA. On the other hand, the presence of a torus is a favourable anatomical feature may lead to a better response to treatment with MADs [10].

In our study, the presence of a mandibular lingual torus was associated with greater hyoid retroposition (*p* = 0.055); in other words, the lower the value of the C3-hyoid distance, the greater the frequency of the presence of mandibular tori and the better the response to treatment with MADs. 

A potential hypothesis underpinning our results to explain the role of torus could be the theory that the mandibular torus is a protective factor in anatomically unfavourable situations, although it is an explanation that does not refute the literature and would require studies directed at the function of the hypoglossus muscle. Thus, patients with a more distal hyoid position would require greater traction on the genioglossus muscle. This would lead to hypertrophy of the mandibular torus, which would act both as a compensation system against the effort that the genioglossus muscle must develop and as a predictive factor for the success of treatment with MADs by providing more favourable traction to the mandibular advancement itself.

In the present study, we also found that current smokers were less likely to have MAD success. Smoking is responsible for chronic gum inflammation, and thus, weakening of the supporting tissue of the teeth [34]. Smoking habits have a detrimental effect on the incidence and progression of periodontitis. This may also be a contraindication treatment with MADs. Therefore, it seems logical to consider the negative effects of smoking on MADs themselves. We could not find any references on the effect of smoking and the success of MADs, and we believe that future studies should also include this variable. However, a recent study showed that smoking negatively influenced MAD tolerance [35]. In our study, the side effects observed in most cases were mild and transient. 

Based on our results, the assessment of the mandibular torus seems to be an indispensable variable in patients undergoing treatment with MADs as a predictor of therapy success. We could not find a variable that was easier to obtain, cheaper, and more predictive of the efficacy of MADs. Revealing the impact of the torus in MAD treatment responses should be the priority of future studies with larger numbers of patients.

## 5. Conclusions

The mandibular torus is an anatomical variable associated with the success of MAD treatment. Its presence practically triples the probability of MAD success. This is the simplest examination with the greatest benefits in terms of the efficacy. In the present study, we did not identify any cephalometric variables that were predictive of treatment success.

## Figures and Tables

**Figure 1 ijerph-19-14154-f001:**
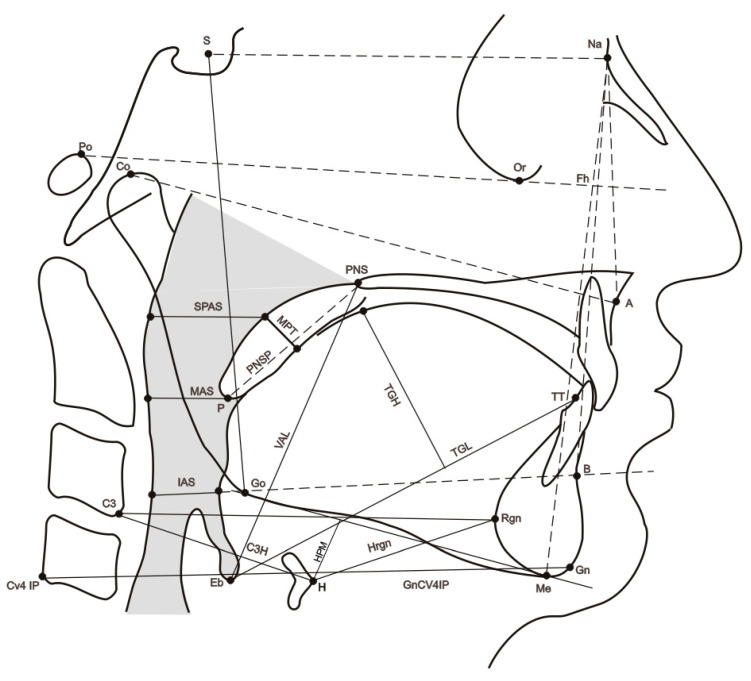
Sagittal cephalometric view showing the reference parameters (points, planes, angles, and Cartesian coordinates) used in the study. Cephalometric points. Na: nasion, most anterior point of the fronto-nasal suture; S: sella, geometric centre of the sella turcica; Eb: base of the epiglottis; H: most anterior point of the body of the hyoid bone; TT: tip of the tongue; RGn: most posterior point of mandibular symphysis at the level of the mid-sagittal plane; CV4IP: most inferior point of the C4; Co: most anterior and superior point of the mandibular condyle; Gn: most anterior and inferior point of the mandible at the level of the mid-sagittal plane of the symphysis; B: most posterior point of the anterior concavity of the mandibular symphysis; Go: most posteroinferior point of the mandible; A: most posterior point of the anterior concavity of the maxillary bone. Me: inferior most point of mandibular symphysis. Po: most superior point of the external auditory canal.

**Figure 2 ijerph-19-14154-f002:**
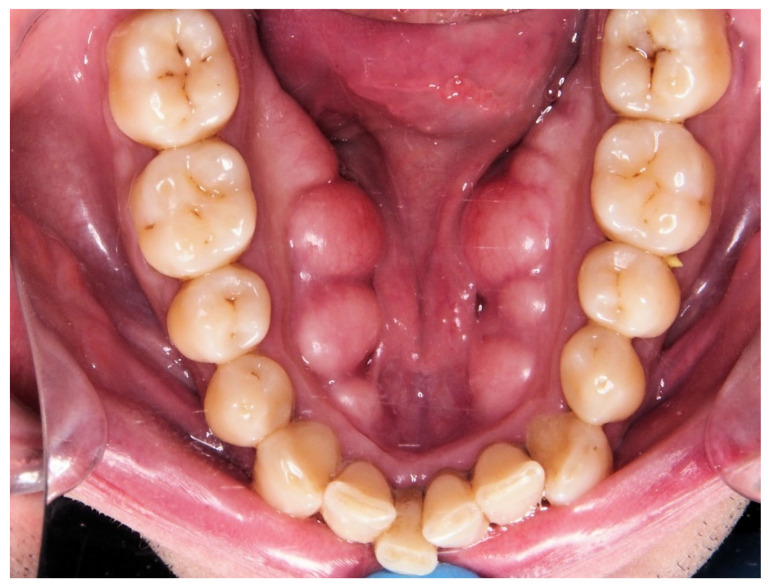
Mandibular torus.

**Figure 3 ijerph-19-14154-f003:**
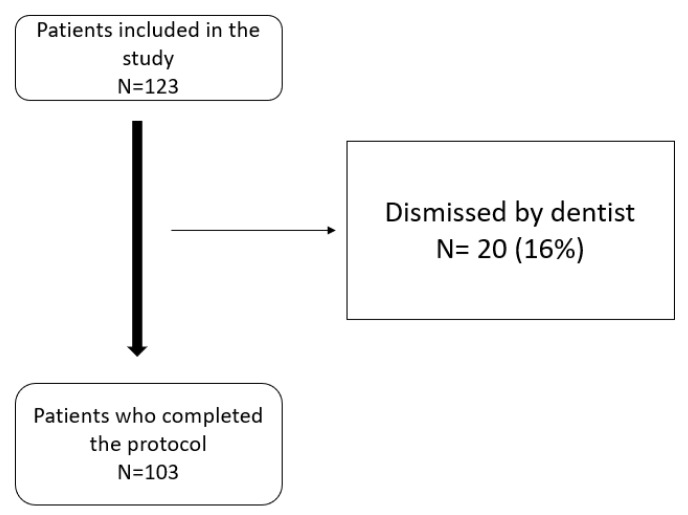
Patient flow chart.

**Figure 4 ijerph-19-14154-f004:**
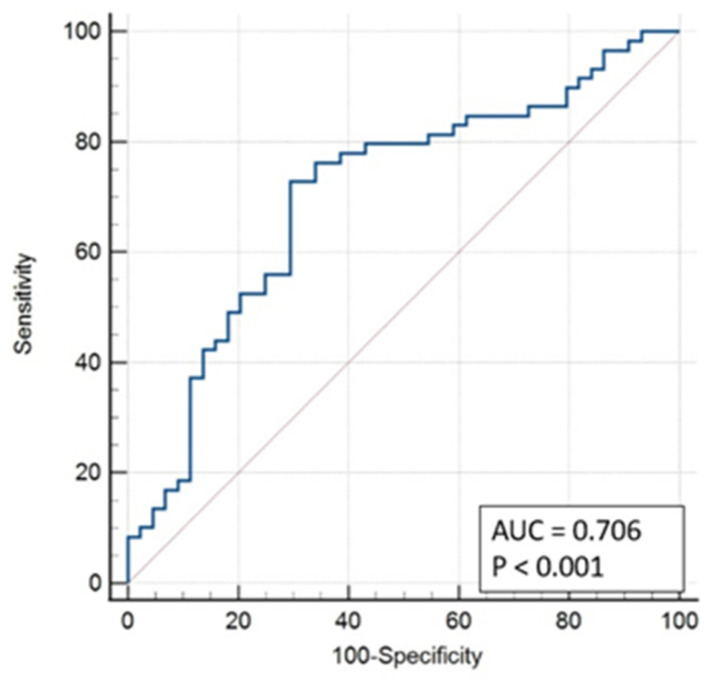
ROC Curve Model 1.

**Figure 5 ijerph-19-14154-f005:**
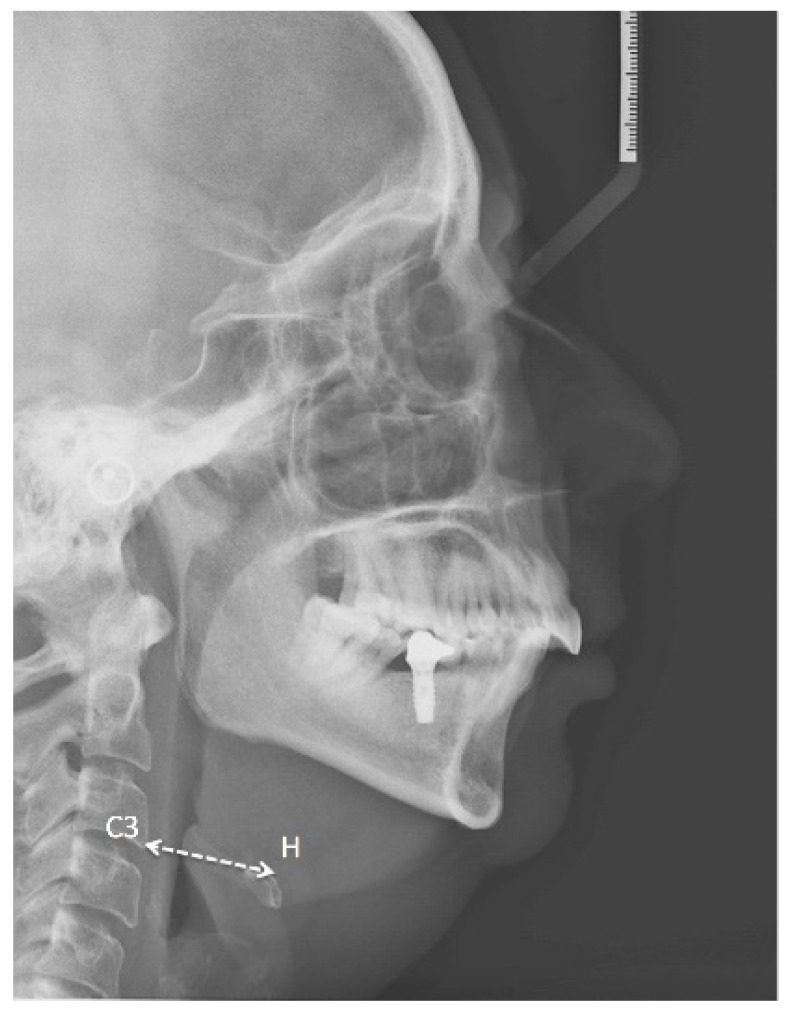
Lateral neck X-ray of one of our patients showing Hyoid-C3 vertebra distance.

**Table 1 ijerph-19-14154-t001:** Cephalometric variables and their abbreviations.

1. SNA	Horizontal Upper Jaw Relationship.
2. SNB	Horizontal mandibular relationship.
3. Midface length	Condilion Point-A. Midface length.
4. Gn-CV4 IP.	Distance from the Gation (Gn) to the most inferior tip of the C4.
5. OCC	Occlusal plane (OCC) to the S-N plane. Angle formed by the occlusal plane to the sella-nasion plane. Vertical maxillomandibular relationship.
6. MPSN	Angle formed by the mandibular plane and the S-N line.
7. Overbite	Distance between the superior and inferior incisal edges measured perpendicularly to the occlusal plane.
8. Overjet	Distance between the upper and lower incisal edges measured in the occlusal plane.
9. Hfaceant	Anterior face height (Na-Me).
10. Hfacepost	Posterior Face Height (S-Go).
11. MPH	HYOID MP PERP Perpendicular to the mandibular plane (PMD) passing through the H point. Distance of the hyoid bone from the lower jaw.
12. HRGn	Hyoid to C3-Retrognathion Line joining point H with point (Retrognation) RGn. Anatomically, this corresponds to the floor of the mouth.
13. C3H	H-C3 Line joining point H with point C3.
14. TGL	Tongue Length (mm). Length of the tongue.
15. TGH	Dorsum of tongue. Maximum height of the dorsum of the tongue.
16. PNSP	PNS to P (see Figure 1). Length of the soft palate.
17. MPT	Maximum soft palate thickness.
18. SPAS	Superior airway space. Measurement (mm) of the airway between the soft palate and the posterior pharyngeal wall along a line parallel to the Go-B plane, passing through the most posterior and superior point of the soft palate.
19. MAS	Middle airway space. Measurement (mm) of the airway between the anterior wall and the posterior pharyngeal wall along a line parallel to the Go-B plane, passing through point P.
20. IAS	Inferior airway space. Measurement (mm) of the airway located between the anterior wall and the posterior pharyngeal wall along the Go-B line.
21. VAL	(PNS-Eb) Oropharyngeal length.

**Table 2 ijerph-19-14154-t002:** (**a**) Characteristics of the 103 patients. (**b**) Characteristics of the patients depending on whether torus is present or not.

(a)			
Variable			
**Age** Mean (SD)		46.3 (9.1)	
**Sex** Nº (%)			
Men		92 (89.3%)	
Women		11 (10.7%)	
**Current smoker** Nº (%)		33 (32%)	
**BMI** Mean (SD)		28.6 (3.6)	
**Epworth** Mean (SD)		11.1 (4.8)	
**AHI pre-MAD** Mean (SD)		31.4 (16.2)	
**AHI post-MAD** Mean (SD)		11.3 (9.2)	
**Torus.** Nº (%)		34 (33%)	
**(b)**			
**Variable**	**Torus**	**No Torus**	** *p* **
**Age** Mean (SD) ^a^	45.15 (8.5)	46.6 (9.5)	0.396
**Sex.** Nº (%) ^b^			0.802
Men	30 (88.2%)	62 (89.9%)	
Women	4 (11.8%)	7 (10.1%)	
**Current smoker** Nº (%) ^b^	9 (26.5%)	24 (34.8%)	0.395
**BMI WHO classification** Nº(%) ^b^			0.176
<25	5 (16.1%)	6 (9.4%)	
25–29.9	20 (64.5%)	34 (53.1%)	
≥30–34.9	6 (19.4%)	24 (37.5%)	
**Epworth** Mean (SD) ^a^	10.6 (5.2)	11.4 (4.7)	0.462
**Supine** Nº (%) ^b^ mean (SD)	18 (54.5%)	29 (45.3%)	0.389
**ODI pre-MAD** Mean (SD) ^a^	22.1 (16.6) (16.7)	26.6 (18.9)	0.231
**AHI pre-MAD** Mean (SD) ^a^ mean (SD)	28.2 (13.1)	32.5 (16.9)	0.250

BMI: Body mass index; WHO: World Health Organization. ^a^ Chi squared test ^b^ Mann–Whitney test.

**Table 3 ijerph-19-14154-t003:** Overall changes in respiratory variables according to the groups (Torus or No Torus).

	Torus			No Torus		
	Pre-MAD Mean (SD)	Post-MAD Mean (SD)	*p*	Pre-MAD Mean (SD)	Post-MAD Mean (SD)	*p*
**AHI**	28.7 (13.4)	7.1 (6.2)	<0.001	33.6 (17.4)	13.6 (9.9)	<0.001
**ODI**	22.1 (17.5)	7.2 (7.6)	<0.001	27.9 (19.5)	11.7 (9.7)	<0.001
**CT90%**	6.5 (9.3)	2.1 (3.7)	0.024	8.5 (14.6)	5.8 (12.3)	0.228
**Mean Sat%**	94.0 (1.6)	94.0 (1.6)	0.312	92.1 (9.7)	93.8 (1.6)	0.148

AHI, Apnoea–Hypopnoea Index; CT90%, percentage of recording time with SaO_2_ values <90%; ODI, oxygen desaturation index (hourly average number of desaturation episodes); mean SaO2, mean arterial oxyhemoglobin saturation.

**Table 4 ijerph-19-14154-t004:** MAD effectiveness.

	% 95%Confidence Interval	
Effective (AHI < 10 with decrease of more than 50% from baseline)	58.3	48.2–67.9
Partially effective (AHI ≥ 10 with a decrease of more than 50% from baseline)	12.6	6.8–20.6
Not effective (AHI decrease lower than 50% from baseline, regardless of the current AHI)	29.1	20.6–38.9

**Table 5 ijerph-19-14154-t005:** Univariate analysis of the OR of the effectiveness of the MAD intervention.

	OR	CI	*p*
Age (years)	0.97	0.930–1.015	0.196
Female sex	0.88	0.251–3.103	0.846
SNA	0.98	0.855–1.085	0.837
SNB	1.02	0.882–1.112	0.302
Midface length	0.98	0.924–1.045	0.628
Gn-CV4 infP	0.83	0.963–1.028	0.997
OCC	0.94	0.887–1.053	0.194
MPSN	0.98	0.934–1.064	0.585
Overbite	1.06	0.844–1.291	0.583
Overjet	0.84	0.637–1.110	0.217
Hfaceant	1.01	0.978–1.062	0.543
Hfacepost	1.03	0.973–1.087	0.319
C3H	0.95	0.902–1.054	0.729
TGL	0.99	0.979–1.082	0.339
TGH	1.05	0.970–1.101	0.375
PNSP	1.04	0.942–1.119	0.358
MPT	1.07	0.814–1.296	0.503
SPAS	1.01	0.894–1.128	0.892
MAS	0.95	0.864–1.128	0.738
IAS	0.99	0.889–1.205	0.543
VAL	1.01	0.966–1.069	0.812
BMI	0.91	0.814–1.024	0.112
Supine-dependent respiratory events	1.50	0.666–3.394	0.327
Smoking history	0.49	0.213–1.140	0.098
Neck circumference	0.92	0.794–1.083	0.343
Torus	3.54	1.409–8.919	0.007

OR, odds ratio; CI, confidence interval 95%. See abbreviations Table 1.

**Table 6 ijerph-19-14154-t006:** Multivariate analysis of the OR of effectiveness of the MAD intervention. Model 1.

	OR	95% CI	*p*
**Age**	0.97	0.924–1.017	0.209
**Sex**	1.39	0.332-5.821	0.653
**BMI**	0.93	0.818- 0.933	0.296
**OCC**	0.97	0.884–0.969	0.506
**Overjet**	0.84	0.628–1.128	0.248
**Current smoker**	0.55	0.222–1.366	0.198
**Torus**	2.85	1.075–7.577	0.035

BMI, body mass index; OR, odds ratio; 95% CI: 95% confidence interval. See abbreviations Table 1.

**Table 7 ijerph-19-14154-t007:** Comparison of cephalometric values according to the presence of a mandibular torus.

	No Torus		Torus		
Variable	Median	P25–P75	Median	P25–P75	*p* *
SNA	81.1	78.5–82.7	81.8	79.5–83.8	0.345
SNB	77.9	75.9–79.8	78.9	75.7–80.7	0.502
midface length	82.8	78.8–88.0	82.0	79.1–86.0	0.680
Gn-CV4 infP	101.6	96.3–106,3	102.2	93.8–108.2	0.683
OCC	17.6	14.2–19.7	14.7	11.6–19.8	0.317
MPSN	35.5	30.0–39.6	33.0	30.2–37.7	0.277
Overbite	1.5	0.4–2.7	2.1	1.1–3.3	0.175
Overjet	3.8	3.0–4.7	3.5	2.9–4.6	0.671
Alfaciala	124.6	116.4–131.2	120.4	114.7–124.3	0.164
Alfacialpost	82.6	77.4–85.9	82.6	76.4–87.4	0.721
C3H	40.2	36.4–43.1	37.5	36.2–40.4	0.055
TGL	81.6	76.6–84.9	80.3	74.9–84.9	0.411
TGH	29.5	25.9–32.6	27.9	25.5–31.4	0.228
PNSP	38.3	35.4–41.3	38.4	34.3–40.7	0.972
MPT	10.7	9.6–12.0	10.2	9.5–11.5	0.409
SPAS	9.2	7.2–12.0	9.0	6.8–11.2	0.818
MAS	12.5	10.0–14.4	11.6	9.4–13.0	0.150
IAS	10.5	9.2–12.7	10.0	7.8–13.6	0.492
VAL	75.0	68.6–81.5	74.7	71.0–78.2	0.662

* Mann–Whitney *U* test. See abbreviations Table 1.

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
