# Peer review of "Mandibular Torus as a New Index of Success for Mandibular Advancement Devices"

_ijerph, 2022, doi:10.3390/ijerph192114154_

Round 1
Reviewer 1 Report
Dear Authors,
thank you for submitting.
Some sentence to correct
Line 19 ... not only ... sleep hygiene, postural management, weight loos
Line 44 improve English
Line 45 passive manipulation ... what do you mean?
Line 81 relevant symptomatology. Please explain better
Line 85 incapacitating somnolence ... what do you mean?
Line 89 temporomandibular articulation alterations. The correct term is temporomandibular disorders (TMD) and they are not a contraindication.
Line 97, 98 All patients underwent an orthopantomography and lateral neck X-ray for subsequent assessment by the orthodontist. Which and why a subsequent assessment? This protocol is not the standard
Levrini L, Sacchi F, Milano F, Polimeni A, Cozza P, Bernkopf E, Segù M; Italian dentist work group about OSAS Collaborators, Zucconi M, Vicini C, Brunello E. Italian recommendations on dental support in the treatment of adult obstructive sleep apnea syndrome (OSAS). Ann Stomatol (Roma). 2016 Feb 12;6(3-4):81-6.
Line 102 Acclimatisation what do you mean? Tritation is the right word
Line 139 Defining MAD effectiveness Reference?
Table 2 AIH pre-MAD correct in AHI
Reviewer 2 Report
Dear Authors, In the reviewers opinion this study, very welll designed and performed, adequately written and presented is worth of publication. Some small editorial remarks are as follows: in the reference list the citations should be presented in homogeneous way (e.g. one position has doi), v.115-117: this sentence should be rather divided in two separate sentences, i.e. separating radiological and physical findings (if possible a little bit more description of what was considered as a torus might be useful for possible inclusion of this findings into clinical practice, as not in all the sleep centers this examinations belongs to routine ), v. 182: kg/m2 should be added to 28.6, v. 315 influenced instead of influences
